# Investigating ethical tradeoffs in crisis standards of care through simulation of ventilator allocation protocols

Jonathan Herington[1,2]*, Jessica Shand[1,3], Jeanne Holden-Wiltse[4], Anthony Corbett[4], Richard Dees[2], Chin-Lin Ching[5], Margie Shaw[1], Xueya Cai[6], Martin Zand[4,7]

1 Department of Health Humanities and Bioethics, University of Rochester, Rochester, New York, United States of America, 2 Department of Philosophy, University of Rochester, Rochester, New York, United States of America, 3 Department of Pediatrics, University of Rochester, Rochester, New York, United States of America, 4 Clinical and Translational Sciences Institute, University of Rochester, Rochester, New York, United States of America, 5 Department of Medicine, University of Rochester, Rochester, New York, United States of America, 6 Department of Biostatistics and Computational Biology, University of Rochester, Rochester, New York, United States of America, 7 Division of Nephrology, Department of Medicine, University of Rochester, Rochester, New York, United States of America

* jonathan.herington@rochester.edu

**Data Availability Statement:** De-identified data is available through the University of Rochester Research Repository at https://doi.org/10.60593/ur.d.25335082.v1. Simulation and analysis code is

## Abstract

### Introduction

Arguments over the appropriate Crisis Standards of Care (CSC) for public health emergencies often assume that there is a tradeoff between saving the most lives, saving the most life-years, and preventing racial disparities. However, these assumptions have rarely been explored empirically. To quantitatively characterize possible ethical tradeoffs, we aimed to simulate the implementation of five proposed CSC protocols for rationing ventilators in the context of the COVID-19 pandemic.

### Methods

A Monte Carlo simulation was used to estimate the number of lives saved and life-years saved by implementing clinical acuity-, comorbidity- and age-based CSC protocols under different shortage conditions. This model was populated with patient data from 3707 adult admissions requiring ventilator support in a New York hospital system between April 2020 and May 2021. To estimate lives and life-years saved by each protocol, we determined survival to discharge and estimated remaining life expectancy for each admission.

### Results

The simulation demonstrated stronger performance for age-sensitive protocols. For a capacity of 1 bed per 2 patients, ranking by age bands saves approximately 29 lives and 3400 life-years per thousand patients. Proposed protocols from New York and Maryland which allocated without considering age saved the fewest lives (~13.2 and 8.5 lives) and life-years (~416 and 420 years). Unlike other protocols, the New York and Maryland algorithms did not generate significant disparities in lives saved and life-years saved between

available at https://github.com/jcherington/triage-sim.

**Funding:** JH, JHW, AC, and MZ received funding through the University of Rochester CTSA award number UL1 TR002001 from the National Center for Advancing Translational Sciences of the National Institutes of Health. The funder played no role in the study design, data collection and analysis, decision to publish or preparation of the manuscript.

**Competing interests:** The authors have declared that no competing interests exist.

White non-Hispanic, Black non-Hispanic, and Hispanic sub-populations. For all protocols, we observed a positive correlation between lives saved and life-years saved, but also between lives saved overall and inequality in the number of lives saved in different race and ethnicity sub-populations.

## Conclusion

While there is significant variance in the number of lives saved and life-years saved, we did not find a tradeoff between saving the most lives and saving the most life-years. Moreover, concerns about racial discrimination in triage protocols require thinking carefully about the tradeoff between enforcing equality of survival rates and maximizing the lives saved in each sub-population.

## 1.0 Introduction

The allocation of scarce resources during public health emergencies is often presumed to require balancing several high-level ethical goals. For scarce resource allocation, potential ethical goals may include (i) saving the most lives [1], (ii) saving the most life-years [2, 3], (iii) respecting principles of non-discrimination [4], or (iv) promoting health equity [5]. Crisis Standard of Care (CSC) protocols, which allocate scarce hospital resources in the setting of public health emergencies, are one such area of policymaking. Published and widely recognized protocols, such as the 2015 New York State (NYS) Ventilator Allocation Guidelines, the 2021 Maryland hospital consortium protocol, and the 2020 Colorado Recommendations for Crisis Standards of Care [1, 6, 7] are taken to encode judgements about how to weight competing ethical values. For instance, the NYS guidelines use the Sequential Organ Failure Assessment (SOFA) with the stated goal of maximizing survival to discharge [1], while the Maryland guidelines use SOFA plus quantitative adjustments for comorbidities to balance short term and long-term survival [7, 8]. These protocols make an implicit assumption that the chosen clinical criteria will promote the intended ethical goal. Moreover, many commentaries [2, 3, 5] assume that tradeoffs exist between goals—for example between saving the most lives and ameliorating health inequities. Because these policies are implemented in complex, dynamic systems, these assumptions may turn out to be false. Modelling of COVID-19 vaccine allocation strategies, for example, suggested no tradeoff between vaccination strategies that saved the most lives and life-years because of the strong association between age and COVID-19 mortality [9]. Rigorous and reproducible testing of ethical assumptions is therefore essential if CSC protocols are to serve their intended ethical goals.

Recent work has attempted to simulate the implementation of CSC protocols in the setting of COVID-related scarcity. Walsh et al. [10, 11] and Chuang et al. [12] simulated the implementation of the New York protocol, but did not compare it to other protocols. Miller et al. [13] and Wunsch et al. [14] assessed the predictive value and potential racial bias of popular protocols, but used a convenience data set of COVID negative individuals without simulating demand or resource scarcity. Bhavani et al. [15] used simulation methods to investigate allocation protocols in the setting of scarcity, but simulated triage only for COVID positive patients and did not estimate post-discharge life expectancy. Finally, Kim et al. [16] used a discrete event simulation approach, with accompanying demand models for COVID patients, to train a bespoke machine learning model for prioritizing critical care beds, but they did not assess

the performance of proposed triage protocols. None of the previously reported simulation methods attempted to evaluate the tradeoff between saving the most lives to discharge and saving the most life-years.

We expand on this work to develop a simulation of CSC protocols that estimates tradeoffs between lives saved, life-years saved, and equal allocation of resources amongst racial groups. We ground our analysis in a more realistic context than prior work by evaluating previously published CSC protocols in the setting of a mixed population of COVID+ and COVID- patients. Our hope is that our application of quantitative simulation to a real-world clinical dataset can be broadly adapted to other bioethical debates about the appropriate mechanism for resource allocation in health systems.

## 2.0 Methods

We sought to simulate the implementation of six proposed ventilator allocation protocols: 1) lottery; 2) age-banding; 3) pure SOFA 4) SOFA bands (New York, 2015), 5) SOFA plus comorbidity (Maryland, 2021), and 6) a different SOFA plus comorbidity model with age as integral consideration (Colorado, 2020) (see Table A in S1 Appendix for an overview). The age-banding protocol assigns priority by 10-year age bands, with ties broken by lottery. The New York '15 protocol assigns priority tiers by SOFA score (SOFA $\leq 7$ = Tier 1; SOFA 8–11 = Tier 2; SOFA $\geq 12$ = Tier 3), with ties broken through a lottery. The Maryland '21 protocol [7] assigns points by SOFA bands ($\leq 8$ SOFA = 1 pt; 9–11 SOFA = 2 pts; 12–14 SOFA = 3 pts; >14 SOFA = 4 pts) and adds +3 points for any patient with a "severe" comorbidity defined as "death likely within 1 year" [7, 8], with ties broken by lottery. Following Bhavani et al. [15], we chose to implement this <1 year mortality estimate using the Elixhauser acute comorbidity summary score (van Walraven variant [17]). We adopted a summary score threshold of $\geq 12$ as our cutoff for "severe" comorbidity, which is the weight assigned to metastatic cancer by van Walraven et al., and the Elixhauser comorbidity with the highest odds ratio of 1 year mortality in multiple studies [18, 19]. The Colorado '20 protocol assigns points by SOFA score band ($\leq 5$ SOFA = 1 pt; 6–9 SOFA = 2 pts; 10–12 SOFA = 3 pts, >12 SOFA = 4 pts), and by a modified Charlson comorbidity index that includes age as a consideration (Table B in S1 Appendix), with ties broken by lottery [20]. Notably, we simulated the December 2020 version of the Colorado protocol [6] that included a traditional SOFA score, but was superseded in 2021 with a modified SOFA score that attempted to reduce racial biases thought to be caused by the Creatinine sub-score [21]. To simplify this study, we ignored protocol instructions that involved re-assessment and re-assignment of ventilators.

Our study population included all critically ill adult patients who received mechanical ventilation between April 2020 and May 2021 within the University of Rochester Medical Center system (including three hospitals in the Rochester metropolitan area). The URMC Coronavirus Ethics Response Group convened in March 2020 to adapt the 2015 NYS Ventilator Allocation Guidelines into a usable triage algorithm [22]. Using a Clinical Translational Science Institute research database and in collaboration with hospital IT, the group created a robust data workflow that allowed for the extraction and database capture of clinical information from the electronic medical record (EMR) system at 30 minute intervals on all inpatient that could then be used as the basis for allocation decisions. For each patient, the data included their ventilator status, their time on ventilator in days, and their SOFA score as calculated by the EMR using the original 1997 definition [23]. The EMR calculated SOFA as the sum of the maximum score of each of the six SOFA sub-scores in the past 24 hours. Missing values relevant to SOFA score calculation were assumed to be normal (rather than imputed), aligning with common practice and the necessity for timely data in a resource shortage [14, 24].

Through a data broker, we received fully de-identified data from the database combined with additional data for COVID infection status, age, sex, self-identified race and ethnicity, acute comorbidities (ICD-10 diagnosis), and discharge status from electronic health records for all subjects. We also received the SOFA score present in the EMR at the point of intubation, which is calculated using data collected in the 24 hours prior to intubation.

Our study involved secondary re-use of a de-identified version of this dataset. It was deemed by the University of Rochester Research Subject Review Board as exempt from review and a waiver of informed consent was granted. The dataset was fully anonymized by an honest broker before being accessed by the study team on the 15th of January 2022.

4604 admissions involving mechanical ventilation, from 4147 unique patients, were identified during the study period. Of these, 897 admissions (19.5%) were excluded because: (i) they lacked a SOFA score associated with their initial intubation (n = 320), (ii) had no discharge disposition, ICD-10 codes or COVID test data (n = 245), or (iii) had multiple intubations within a single admission, but no indication of which SOFA score was associated with initial intubation and which was associated with re-intubation after trial extubation (n = 332). We decided to remove this last set of encounters from the simulation dataset in order to ensure that each allocation decision concerned an intubation rather than a mix of initial intubations and re-intubations. The distribution of age, sex, race and COVID positivity amongst the excluded patients was similar to the included patients. There was no significant difference in survival rates amongst the excluded (.72 [.71-.74]) and included (.72 [.69-.76]) encounters (Table A in S2 Appendix). Demographic features of the final modelling cohort of 3707 encounters from 3512 unique patients are described in Table 1 and in S2 Appendix.

We simulated the implementation of each protocol under different levels of scarcity using a Monte Carlo method. Extending the approach of Bhavani et al., [15] our model simulates an $n/20$ (i.e. 5%, 10%...95% capacity) ventilator shortage by (1) randomly sampling twenty patients <A, B, . . .., T> from our dataset, and assigning each patient to one of ten decision "pairs" randomly numbered 1 through 10 (e.g. 1:[A, R], 2:[H, C]. . ..10: [Q, L]), (2) ranking the patients in each pair for priority based on the relevant protocol (e.g. 1:[A > R], 2: [C > H]. . ..10:[Q>L]), (3) allocating beds to both patients in the first $n$-10 pairs (i.e. if n = 15, then assign [A = 1, R = 1], [C = 1, H = 1] etc. to first five pairs), and not allocating beds to either patient in the last 10-$n$ pairs (i.e. if n = 15, then no pairs miss out on beds), 4) allocating a bed to the highest priority patients in each remaining pair (e.g. [Q = 1, L = 0]), and 4) repeating this process until all patients in our dataset have been allocated (or not) a bed. We estimated survival in each simulation by observing the actual (scarcity-free) survival to discharge of patients assigned beds. We assumed that patients who were not allocated a bed in our simulation did not survive. We repeated the simulation 1,000 times for the 50% capacity results and 250 times for all other capacity levels and sensitivity analyses.

For each simulation, we calculated (i) survival rate, (ii) age-adjusted survival rate, and (iii) aggregate comorbidity-adjusted life expectancy (in years). Due to COVID's severe age-associated mortality and the distribution of age within racial groups, survival rates for sub-populations were age-adjusted, and confidence intervals for both raw and age-adjusted rates were calculated using the modified Gamma method of Tiwari et al. [25, 26] Raw life expectancy was calculated for each subject from the corresponding National Vital Statistics System (NVSS) life tables for their age, sex and race. The impact of comorbidities on life expectancy was estimated by applying the adjustments previously calculated by Cho et al. for Medicare recipients without a history of cancer [27]. Subjects were placed into one of three comorbidity bands (none, low/medium, high) using the weights identified by Cho, and the corresponding age adjustment in Cho et al. was applied, before re-calculating the subject's life expectancy using the NVSS tables.

**Table 1. Descriptive statistics for mechanically ventilated adult patients, March 2020 to April 2021.**

| Feature | n | % | Survival | Age-Adjusted Survival |
|---|---|---|---|---|
| | | | % (CI) | % (CI) |
| **Overall** | **3707** | | **72 (71–74)** | **84 (82–86)** |
| **Sex** | | | | |
| Male | 2197 | 59 | 73 (71–75) | 84 (80–87) |
| Female | 1510 | 41 | 72 (69–74) | 84 (80–86) |
| **Age** | | | | |
| <25 | 104 | 03 | 93 (86–97) | |
| 25–34 | 235 | 06 | 90 (85–93) | |
| 35–44 | 247 | 07 | 83 (77–87) | |
| 45–54 | 456 | 12 | 81 (77–85) | |
| 55–64 | 868 | 23 | 76 (73–79) | |
| 65–74 | 980 | 26 | 69 (66–73) | |
| 75–84 | 629 | 17 | 60 (54–64) | |
| >85 | 188 | 05 | 50 (38–59) | |
| **Race** | | | | |
| AAPI, non-Hispanic | 50 | 01 | 78 (61–89) | 87 (24–97) |
| AIAN, non-Hispanic | 3 | >01 | 1.0 | 1.0 |
| Black, non-Hispanic | 577 | 16 | 76 (72–80) | 82 (74–87) |
| Hispanic | 152 | 04 | 82 (73–88) | 89 (78–94) |
| White, non-Hispanic | 2752 | 74 | 71 (69–74) | 85 (81–87) |
| >1 Race, non-Hispanic | 16 | >01 | 75 (36–93) | 87 (23–96) |
| Unknown, non-Hispanic | 157 | 04 | | 79 (81–87) |
| **COVID Status** | | | | |
| Negative | 2431 | 66 | 73 (71–75) | 84 (81–86) |
| Positive | 1276 | 34 | 71 (68–74) | 83 (76–88) |
| **SOFA Band** | | | | |
| 1–7 | 3192 | 86 | 76 (74–77) | 87 (84–89) |
| 8–11 | 372 | 10 | 56 (49–63) | 58 (15–80) |
| 11–24 | 143 | 04 | 48 (34–59) | 75 (00–83) |

The comorbidity adjustment for a 65yr old in Cho et al. was applied to all subjects in our cohort <65yrs. We explore the limitations of this approach in the discussion.

We then defined "lives saved" and "years of life saved" per patient for each protocol, $p$, and beds-per-patient capacity level, $(0 < c < 1)$, by first identifying the survival rate, $S_B$, and expected years of life lived post-discharge per patient, $LY_B$, of our population in the baseline no scarcity scenario. Next, for each capacity level, we defined the expected survival rate ($S_B \times c$) and expected years of life per patient ($LY_B \times c$) as the product of the baseline rates and the relevant capacity constraint (e.g. 0.5 beds per patient). Lives saved per patient, $LS_{p,c}$, is the difference between the simulated survival rate for the protocol and the expected survival rate for that capacity. Years of life saved per patient, $LYS_{p,c}$, is the difference between the simulated number of life-years lived per patient for that protocol, and the expected number of life-years lived for that capacity.

All simulation and data analysis was performed in Python v3.11 using the pandas, matplotlib, numpy, scipy, seaborn and stats models packages. De-identified data is available through the University of Rochester Research Repository at https://doi.org/10.60593/ur.d.25335082.v1. Simulation and analysis code is available at https://github.com/jcherington/triage-sim.

## 3. Results

Our population of 3707 ventilator encounters with 3512 unique subjects had a mean (± std) age of 62 (±16) (Table 1). Among them, Black and Hispanic patients were younger (56±17 and 53±18) compared to non-Hispanic White patients (64±15). Black patients were more likely (rate [95% CI]) to be diagnosed with COVID (.45 [.41-.49]), compared to Hispanic (.39, [.32-.47]) and non-Hispanic White patients (.33, [.31-.34]). In addition, SOFA scores at the initial point of intubation were higher for White subjects (mean ± std: 3.69 ± 0.07) than for both Black and Hispanic subjects (3.46 ± 0.15 and 3.14 ± 0.29).

Contrary to prior reported results [5, 14, 15] in our baseline population, we did not observe significant racial disparities in age-adjusted survival amongst White, Black and Hispanic patients who received mechanical ventilation (Table 1). Neither did we observe significant disparities in age-adjusted survival (rate [95% CI]) for patients who were COVID positive (.84 [.81-.87]) and negative (.83 [.76-.88]). In a multiple variable logistic regression model, SOFA score at the point of initial intubation (OR: .896, [.878-.916]), age in years (OR: .964, [.958-.969]), and Elixhauser score (OR: 1.054, [1.033–1.074]) were all predictive of survival, whereas sex, race and COVID-positivity were not significantly predictive (Table A in S3 Appendix).

### 3.1 Allocation inequalities

When simulating a shortage of ventilators where only one bed is available for two patients (c = 0.5), and without controlling for other factors (i.e. clinical acuity), all protocols except Lottery exhibited significant racial disparities in allocation of ventilators (Table 2). Contrary to prior findings with simulated populations of COVID positive patients [15], in our simulations with mixed populations of COVID positive and negative patients, allocation of ventilators favored non-Hispanic Black and Hispanic patients when compared to non-Hispanic Whites. Allocation disparities were most pronounced for the Age-based protocol and least pronounced in the context of the New York protocol.

**Table 2. Allocation and survival by protocol and race and ethnicity.**

| Protocol | Overall Survival (%) | Allocation Rate by Race & Ethnicity (%) | | | Age-Adjusted Survival Rate (%) | | |
|---|---|---|---|---|---|---|---|
| | | Non-Hispanic, Black | Hispanic, All Races | Non-Hispanic, White | Non-Hispanic, Black | Hispanic, All Races | Non-Hispanic, White |
| **Baseline** (No scarcity) | 72.57 (70.83–74.23) | 100 | 100 | 100 | 82.3 (74.5–87.4) | 89.1 (78.4–94.1) | 85 (81.4–87.4) |
| **Lottery** | 36.28 (36.25–36.30) | 50.1 (50.0–50.2) | 50.0 (49.8–50.3) | 50.0 (49.9–50.0) | 41.2 (41.0–41.4) | **44.7** (44.3–45.0) | 42.5 (42.3–42.6) |
| **Age** | **39.19** (39.17–39.21) | **59.8** (59.7–59.9) | **64.8** (64.6–64.9) | 46.6 (46.6–46.6) | 68.2 (68.2–68.3) | **73.7** (73.6–73.9) | 71.1 (71.1–71.2) |
| **Pure SOFA** | 37.95 (37.93–37.96) | **51.8** (51.7–51.9) | **55.2** (55.0–55.4) | 49.7 (49.7–49.7) | **51.3** (51.1–51.4) | 50.3 (50.0–50.6) | 48.4 (48.3–48.6) |
| **New York '15** | 37.590 (37.57–37.61) | **50.4** (50.3–50.5) | **51.1** (50.8–51.3) | 50.0 (50.0–50.1) | 44.5 (44.3–44.7) | **47.2** (46.9–47.5) | 44.4 (44.2–44.6) |
| **Maryland '21** | 37.13 (37.11–37.15) | **50.8** (50.7–50.9) | **51.1** (50.8–51.3) | 50.0 (50.0–50.0) | 44.4 (44.2–44.6) | **47.7** (47.4–48.0) | 43.7 (43.5–43.8) |
| **Colorado '20** | 38.91 (38.89–38.93) | **56.5** (56.4–56.6) | **60.13** (61.1–61.5) | 47.8 (47.8–47.9) | **63.2** (63.1–63.3) | 64.5 (64.2–64.7) | 61.1 (61.0–61.2) |

*Mean overall survival rate and 95% CIs are reported for 1000 Monte Carlo simulations at 50% scarcity.

## 3.2 Survival and lives saved

In a simulation of moderate scarcity (0.5 beds per patient), the Age-based protocol had significantly higher overall survival rates (.3919 [.3917-.3921]) than any other protocol (Table 2). On the other hand, the Maryland (.3713 [.3711-.3715]) protocol resulted in lower overall survival rates than all other protocols except Lottery (.3628 [.3725-.3630]). The Maryland, Colorado, and Pure SOFA protocols exhibited significantly lower rates of age-adjusted survival for White patients compared to both their Black and Hispanic counterparts, with the biggest racial inequalities in age-adjusted survival occurring for the Colorado protocol (White: .611, Black: .632). The Age-based and Lottery protocols exhibited significantly lower age-adjusted survival for Black patients, compared to the White or Hispanic population. Unsurprisingly, the Age-based protocol heavily skewed survival towards younger Age groups, while the New York and Lottery resulted in roughly similar distributions of survival across age-groups (S4 Appendix).

Because survival rates are heavily influenced by the distribution of survival amongst sub-populations in our underlying cohort, we also estimate the number of lives saved by each protocol: i.e. the additional number of lives saved by employing a protocol instead of using a lottery (Table 3 and Fig 1). In a simulation of 0.5 beds per patient, the Age-based (29.1 [28.6–28.9]) and Colorado (26.3 [26.1–26.4]) protocols saved significantly more lives per thousand patients than the nearest age insensitive protocol, Pure SOFA (16.7 [16.5–16.9]). The Maryland (8.5 [8.3–8.7]) and New York (13.1[12.9–13.3]) protocols saved the least lives per thousand patients, both overall and for the Black (Maryland: 13.6 [12.7–14.6]) and Hispanic (New York: 11.0 [8.9–13.1]) sub-populations. This effect was reversed for the White sub-population, for whom the New York protocol saved the most lives (13.5, [13.2–13.8]) and the Age protocol the least (4.4 [4.1–4.7]). All protocols except New York exhibited significantly lower numbers of lives saved for White patients compared to both their Black and Hispanic counterparts, with the biggest racial inequalities in lives saved occurring for the Age and Colorado protocols (Table 3).

After simulating different capacity shortages, we noted that the magnitude of the dominance of the Age and Colorado protocols was sensitive to the degree of scarcity (Fig 2). At 0.5 beds per patient the Age based protocol saves approximately 29 additional lives per 1000 patients, while at 0.9 beds/patient, the protocol saves only 5 lives per 1000 patients (Table B in S5 Appendix). At 0.9 beds per patient, the differences between New York and Pure SOFA, and the Colorado and Age protocols are not statistically significant.

## 3.3 Life-years saved

With respect to life-years saved, for a scarcity level of 0.5 beds per patient, the Age-based protocol saved significantly more life-years per thousand patients (3419 [3414–3425]) than the nearest competing protocol, Colorado (2681 [2674–2687]) (Table 4 and Fig 3). The New York protocol saved fewer life-years than all other protocols, both overall (416 [407–424]) and for

**Table 3. Lives saved by protocol and ethnicity/race at 50% capacity.**

| Protocol | Lives Saved per 1000 patients (95% CI) | | | |
|---|---|---|---|---|
| | Non-Hispanic, Black | Hispanic, All Races | Non-Hispanic, White | Overall |
| **Age** | 101.6 (100.8–102.5) | 144.2 (142.6–145.8) | 4.4 (4.1–4.7) | 29.1 (28.9–29.3) |
| **Pure SOFA** | 41.1 (40.3–42.0) | 44.8 (43.0–46.7) | 12.4 (12.1–12.7) | 16.6 (16.5–16.8) |
| **New York, '15** | 16.1 (15.1–17.1) | 11.0 (8.9–13.1) | 13.5 (13.2–13.8) | 13.1 (12.9–13.3) |
| **Maryland, '21** | 13.6 (12.7–14.6) | 13.6 (11.5–15.7) | 8.1 (7.8–8.5) | 8.5 (8.3–8.7) |
| **Colorado, '20** | 86.0 (85.1–86.8) | 109.2 (107.4–111.1) | 7.7 (7.4–8.0) | 26.3 (26.1–26.4) |

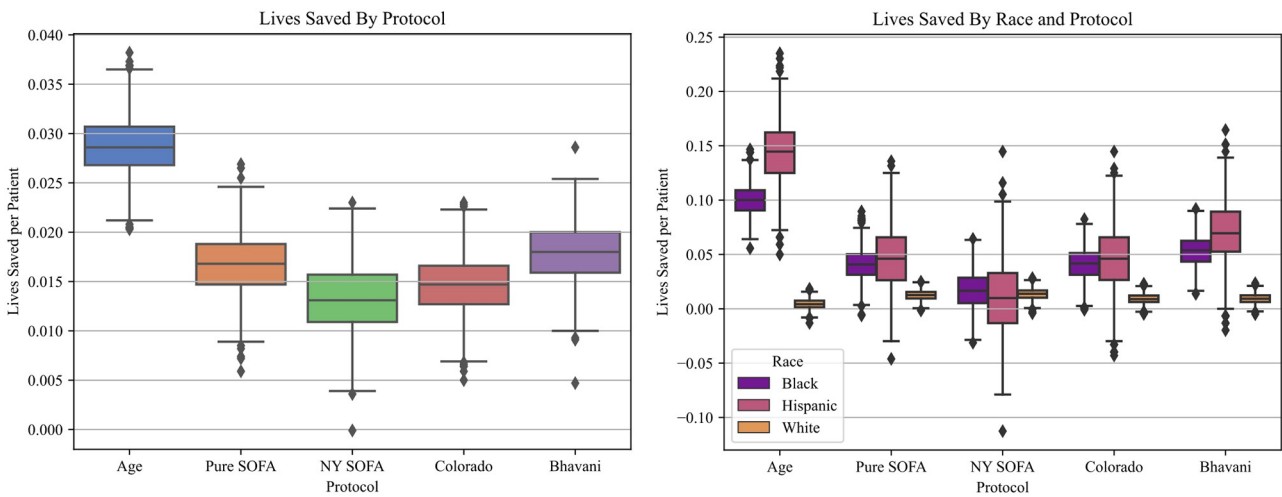

**Fig 1.** Lives saved per patient by protocol: **A**. Overall increase in lives saved per patient, for each protocol, at 50% scarcity. **B**. Lives saved per patient by race/ethnicity, for each protocol, at 50% scarcity.

the Hispanic (450 [366–534]) and Black (626 [591–661]) sub-populations. All five non-random protocols resulted in significantly fewer life-years saved for White patients when compared to both their Black and Hispanic counterparts (Table 4 and Fig 3). After simulating different capacity shortages, we again noted that the difference between Age and other

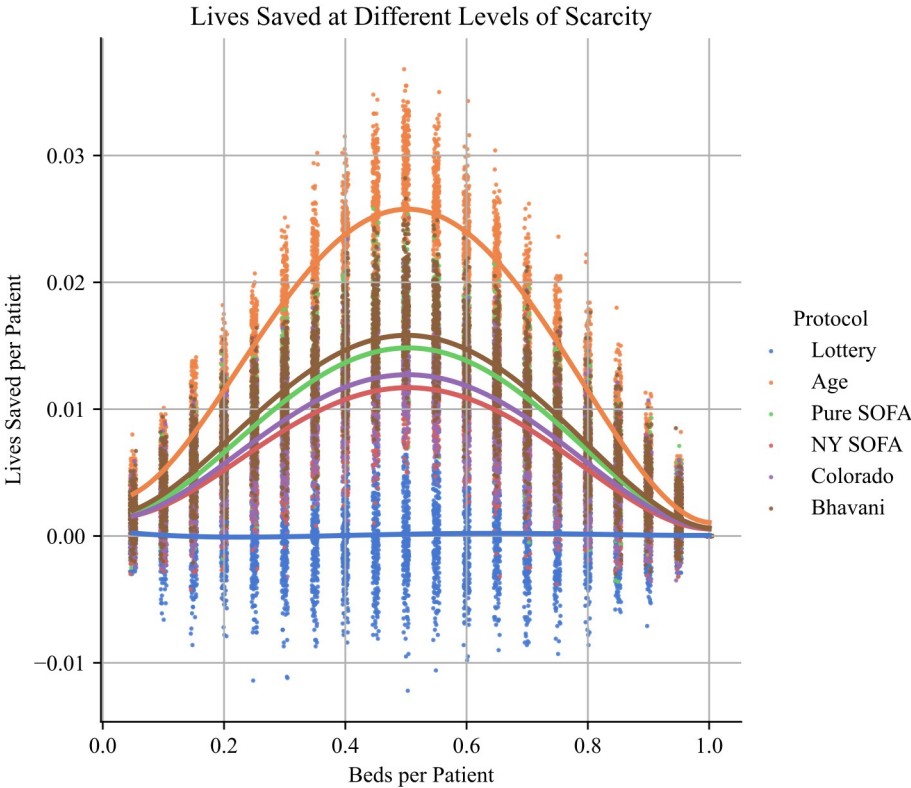

**Fig 2. A. Lives saved at different levels of scarcity, for each protocol.** The greatest differences between protocols occur at moderate levels of scarcity (i.e. ~0.5 beds per patient), and differences between protocols decline at both high and low levels of scarcity.

**Table 4. Life-years saved by protocol and ethnicity/race at 50% capacity.**

| Protocol | Life-years Saved per 1000 patients (95% CI) | | | |
|---|---|---|---|---|
| | **Non-Hispanic, Black** | **Hispanic, All Races** | **Non-Hispanic, White** | **Overall** |
| **Age** | 5556 (5534–5578) | 8581 (8532–8630) | 2557 (2549–2565) | 3419 (3414–3425) |
| **Pure SOFA** | 1958 (1930–1986) | 2188 (2115–2260) | 593 (583–602) | 842 (835–849) |
| **New York, '15** | 626 (591–660) | 441 (361–522) | 368 (357–379) | 409 (400–417) |
| **Maryland, '21** | 682 (648–716) | 773 (689–857) | 354 (343–364) | 423414–432) |
| **Colorado, '20** | 4731 (4707–4755) | 6228 (6164–6291) | 1969 (1961–1978) | 2681 (2674–2687) |

protocols was sensitive to the degree of scarcity, but less so than lives saved. For instance, at 0.9 beds/patient, the Age protocol saves only 682 [675–688] life-years per thousand patients, but continues to outperform the best age insensitive protocol, Pure SOFA (163 [154–173]), by a factor of three (Table C in S5 Appendix).

## 4.0 Discussion

In this simulation study, we found that age-sensitive protocols significantly outperformed all other protocols with respect to both lives saved and life-years saved. At moderate levels of scarcity (i.e. 0.5 beds per patient), we estimate that selecting Age over Lottery could save approximately 29 lives per thousand patients requiring ventilation. When scaled to a national level, this level of effectiveness compares very favorably to classic public health interventions such as seatbelt use (14,955 lives saved per year) [28] or flu vaccination (7,200 lives saved from 36 million infections in 2020, or 0.2 lives saved per 1000 infections) [29]. Because age-sensitive protocols discriminate on the basis of age, they have an uncertain legal status. Among the other three protocols, the difference between the best age insensitive protocol (Pure SOFA) and worst performing (Maryland) is less pronounced but still significant, saving approximately 8 lives and 410 life-years per thousand patients at moderate levels of scarcity.

Notably, the level of scarcity appears to have a dramatic influence on the effect size of choosing different protocols. While the Age and Colorado protocols dominate the other

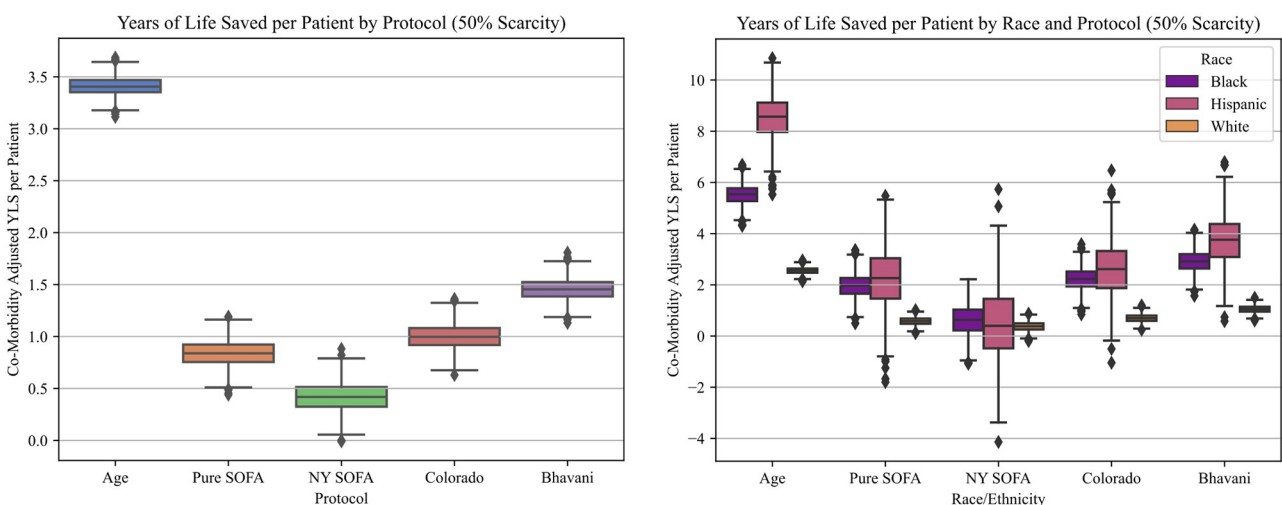

**Fig 3. Life years saved per patient. A.** Overall increase in life years saved per patient, for each protocol, at 50% scarcity. **B.** Life years saved per patient by racial group, for each protocol, at 50% scarcity.

protocols for both severe (0–0.2 beds per patient), moderate (0.4–0.6 beds per patient) and low levels of scarcity, the size of the effect diminishes substantially at low and severe scarcity. At very low levels of scarcity (e.g. 0.9 beds per patient), choosing Age over New York saves 2.9 lives per 1000 patients. The much smaller differences between protocols at that level of scarcity suggest that other ethical considerations such as complexity of implementation, transparency, and social acceptability may be decisive.

One possible explanation for the poor performance of the New York and Maryland protocols is that they have many fewer prioritization bands (i.e. only 3 and 4) than the other protocols (Age = 8 bands, Colorado = 10 bands, Pure SOFA = 23 bands). In principle, more bands add greater discriminatory power to a protocol and so enhance its ability to save lives at the margins. This is borne out by the significant improvement offered by the Pure SOFA protocol over the New York protocol. This empirical result suggests that the value of treating clinically similar cases alike (by having relatively "wide" priority bands) must be weighed against the value of maximizing the overall numbers of lives saved.

### 4.1 Tradeoffs between lives and life-years

Much prior work has assumed that there is an empirical tradeoff between lives saved and life-years saved, thus necessitating a difficult normative decision about prioritizing different values in the context of disagreement [2]. In our simulations, that tradeoff is not evident. Not only does the Age-based protocol save the most lives, it also saves the most life-years. Indeed, this result appears to be robust across the protocols simulated (Fig 4). While there is a high degree of variance in both survival and longevity within each protocol, protocols which save more lives on average also save more life-years on average.

This result call into question the assumed tension between maximizing survival to discharge and maximizing life-years saved, but it also illuminates the different justifications one might have for focusing on the distinction between lives and life-years. For utilitarians, who aim to maximize the aggregate good, this result suggests that the distinction between lives and life-years is empirically unimportant when choosing a pandemic policy. For those who hold a "fair innings" view for the importance of allocating ventilators to younger patients, this result is less important–since saving people with long life expectancies is an ancillary result of prioritizing patients who have yet to progress through the different stages of life. Alongside the performance of the Age protocol, this result shows that prioritizing the young is at least compatible with maximizing the number of lives saved.

### 4.2 Tradeoffs between performance and racial disparities

Of those protocols examined, the protocols that offered the most equal distribution of survival rates over the three racial groups had the least favorable overall performance. In our simulations, at all levels of scarcity and for all protocols except the New York and Lottery, the White sub-population experienced significantly lower allocation rates, lives saved, and life-years saved. This finding is in *prima facie* tension with previously reported simulations of triage protocols with COVID-19 positive cohorts [15], pre-pandemic cohorts [13, 14], and with the lived experience of actual COVID mortality in intensive care units [5, 30]. Age-adjusted rates of survival remained significantly lower for White patients, and thus it is unlikely that the observed racial disparities are solely explained by the different age distributions amongst White non-Hispanic, Hispanic and Black non-Hispanic patients in our sample. Rather, we suspect that hospital-level population bias may explain this difference. In particular, there are two major hospital systems that offer adult critical care services in our study cohort's metropolitan area, and we cannot assume that the distribution of patients between these systems is the same for

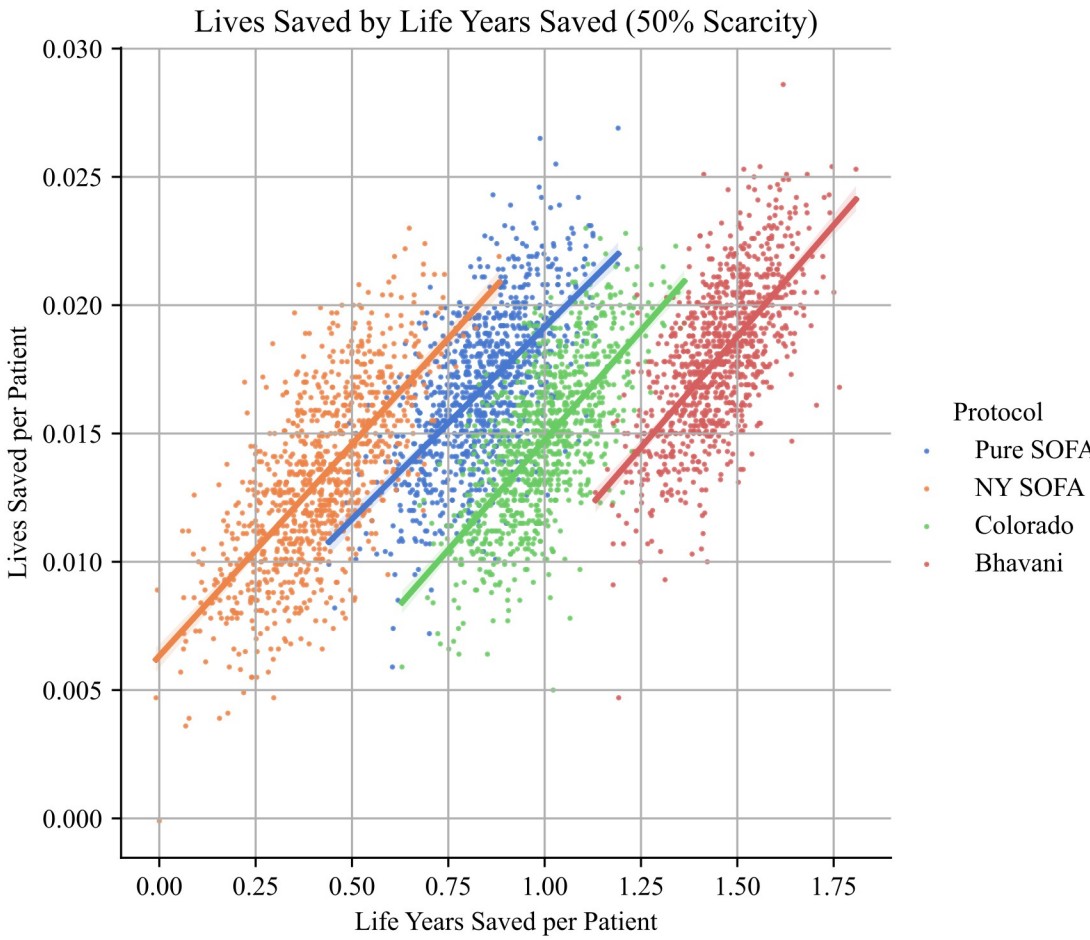

**Fig 4. Tradeoffs between life years saved and lives saved.** While there is significant variance in numbers of lives and life years saved, there is a strong positive correlation between each statistic for all protocols.

each sub-population. Hospitals serve particular patient populations, not national averages. This highlights that implementation of protocols may result in different aggregate outcomes in different contexts (i.e. with national-level results coming apart from regional or hospital level outcomes).

More interestingly, our results suggest a tradeoff between equalizing lives saved across racial groups and maximizing the number of lives saved for each racial group. In our results, protocols other than New York greatly improve the number of lives saved for Black and Hispanic populations while making only marginal differences to the number of lives saved for the White population. Thus, while New York secures equality in survival rates, it would deprive Black and Hispanic populations of very large gains for little benefit in lives or life-years saved for White patients. Choosing New York in such cases would thus be a classic case of 'levelling down'–a situation where permitting inequality would make little difference to the absolute welfare of the worst off group, but could markedly improve the welfare of other groups [31]. Notably, this tradeoff is only observable by simulating the set of alternative policies in a single population, and hence discussion of the value of equality in survival rates benefits from the use of simulation methods.

### 4.3 Limitations

This study has a number of important limitations. The first is that our dataset is limited to a single hospital system in a region with two major healthcare institutions. As we note above, this may partially explain our surprising findings regarding racial disparities in survival, because Black and Hispanic patients with poorer social determinants of health may have been more likely to have been admitted elsewhere. In future research, we hope to extract a region-wide dataset that eliminates hospital-level effects.

Another limitation is that we did not collect, nor adjust for, the existence of a "do-not-resuscitate" order or a decision to undertake compassionate withdrawal. In principle, some patients subject to DNRs may have otherwise survived their admission and, given well characterized racial differences in the uptake of DNR orders, this may have artificially depressed survival rates for White patients relative to other racial sub-populations [32]. While this is true for survival rates, it is less likely to distort differences in the lives saved and life-years saved statistics since it accounts for variation in actual mortality (including withdrawal of treatment and natural expiration) and estimates deviations from "expected" survival rates for each population.

A further limitation is that the patient population during a crisis surge would likely differ from the population within our dataset (collected between April 2020 and May 2021). For example, we would expect a surge to involve the cancellation of elective procedures (e.g. transplant, non-emergent cardiac procedures), fewer trauma cases, and thus a much higher proportion of COVID+ patients. In order to test this assumption, we performed a sensitivity analysis (S6 Appendix) which simulated different mixes of COVID+ and COVID- patients. While there is some statistically significant variation in mean lives saved and/or life years saved by patient mix for the pure SOFA protocol, no other protocol exhibited significant differences by patient mix. An important limitation of this sub-analysis is that we were unable to identify patients who received critical care support because of an elective procedure, and hence we encourage further work to explore this consideration.

A final limitation of our simulation is that it does not allow for "re-allocation" of ventilators. Most existing protocols (including New York's, Maryland's and Colorado's) allow patients a "timed trial" of ventilation, and contemplate mechanisms for removing ventilators from patients who are not improving at specified re-assessment timepoints [33]. The criteria for re-allocation (and the timing of re-assessments) must therefore strike a balance between giving each individual a chance to recover and preventing "wastage" of ventilators on patients who will not recover [34]. In this respect, the criteria for reallocation may have significant effects on the survival of individual patients and the overall numbers of lives saved. Unfortunately, our existing dataset and simulation strategy cannot accommodate reallocation of ventilators, since it only includes SOFA scores collected at point of intubation, and thus envisages allocation until survival or natural expiration. Future work should incorporate time-series datasets of patient trajectories while intubated, and simulation designs that allow for "pools" of ventilators to be allocated and re-allocated. We leave this extension for future work.

### 5.0 Conclusion

In this paper we have empirically tested a number of assumptions that have been at the forefront of bioethical debates over the appropriate CSC protocols. In our simulation using actual patient data, we show several findings. First, there are clear performance differences between protocols, and age-sensitive protocols appear to save more lives to discharge than protocols which rely on SOFA or comorbidity alone. Second, using comorbidity-adjusted estimates of post-discharge life expectancy, we show that there is unlikely to be a tradeoff between saving

lives and saving life-years in the aggregate. Third, we identify that while there was tradeoff between equalizing lives saved between sub-populations and overall performance, those protocols with high levels of inequality were sometimes better for the least racial groups than protocols with more equal distributions of lives saved. These three findings cut to the heart of important bioethical debates, and should inform both philosophical and implementation work to improve CSC policy.

Moreover, we have reproduced and extended a methodology for prospectively and retrospectively analyzing the performance of different CSC protocols, under different levels of scarcity. Building on earlier work by Bhavani et al., we show not only that these simulation methods can be used to estimate survival, but also that they can also provide information about estimated life expectancy, inequalities between sub-populations, and the optimal parameterization of models (i.e. numbers of bands, precise threshold for clinical acuity scores). Nonetheless, the work of empirically exploring CSC protocols is not complete. So far, no simulation protocol has been generated that includes (i) all critical care patients, (ii) is sensitive to duration of resource use, and (iii) accommodates re-allocation of resources. These components are critical to many currently-existing CSC protocols, and they may yield important empirical results that alter or contradict the results from the static allocation simulations conducted here and elsewhere [15]. Without a generalizable methodology for either post hoc or ex ante testing of the ethical assumptions underpinning re-allocation of resources, we will continue to lack evidence that future allocation policies achieve their intended ethical goals. We hope that the preliminary work we present in this paper spurs others to develop more robust, dynamic models that can inform CSC policy choices for the next public health emergency.

## Supporting information

**S1 Appendix. Crisis standards of care protocols.** Tabular summary of simulated protocols. (DOCX)

**S2 Appendix. Characteristics for included and excluded encounters.** (DOCX)

**S3 Appendix. Association between survival, SOFA score and age.** (DOCX)

**S4 Appendix. Survival by age and protocol.** (DOCX)

**S5 Appendix. Capacity sensitivity analysis.** (DOCX)

**S6 Appendix. Patient mix sensitivity analysis.** (DOCX)

## Acknowledgments

The authors would like to acknowledge the efforts of Data Science Masters' students—Derek Caramella, Ezgi Siir Kibris, Nefle Nesli Oruc, and Walter Burnett—for assisting with initial exploration of the dataset under the supervision of JH. Final data acquisition, cleaning, analysis and simulation work was conducted by JH. The authors would like to thank Michael Nabozny, William Parker, Lainie Ross and the members of the URMC African American Patient and Family Advisory Council for comments on earlier versions of the project.

## Author Contributions

**Conceptualization:** Jonathan Herington, Jessica Shand, Richard Dees, Chin-Lin Ching, Margie Shaw, Xueya Cai, Martin Zand.

**Data curation:** Jonathan Herington, Jeanne Holden-Wiltse, Anthony Corbett.

**Formal analysis:** Jonathan Herington.

**Funding acquisition:** Martin Zand.

**Investigation:** Jonathan Herington, Chin-Lin Ching.

**Methodology:** Jonathan Herington, Jessica Shand, Xueya Cai.

**Project administration:** Jessica Shand.

**Resources:** Martin Zand.

**Software:** Jonathan Herington.

**Validation:** Jonathan Herington.

**Visualization:** Jonathan Herington.

**Writing – original draft:** Jonathan Herington.

**Writing – review & editing:** Jonathan Herington, Jessica Shand, Jeanne Holden-Wiltse, Anthony Corbett, Richard Dees, Chin-Lin Ching, Margie Shaw, Xueya Cai, Martin Zand.

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
