## [Decision Letter · Decision Letter 0]

29 Apr 2024

PONE-D-24-08864Investigating Ethical Tradeoffs in Crisis Standards of Care through Simulation of Ventilator Allocation ProtocolsPLOS ONE

Dear Dr. Herington,

Thank you for submitting your manuscript to PLOS ONE. After careful consideration, we feel that it has merit but does not fully meet PLOS ONE’s publication criteria as it currently stands. Therefore, we invite you to submit a revised version of the manuscript that addresses the points raised during the review process.

We look forward to receiving your revised manuscript.

Kind regards,

Ronaldo Go, MD

Academic Editor

PLOS ONE

Journal Requirements:

"The authors would like to acknowledge the efforts of Data Science Masters’ students - Derek Caramella, Ezgi Siir Kibris, Nefle Nesli Oruc, and Walter Burnett - for assisting with initial exploration of the dataset under the supervision of JH. Final data acquisition, cleaning, analysis and simulation work was conducted by JH. The authors would like to thank Michael Nabozny, William Parker, Lainie Ross and the members of the URMC African American Patient and Family Advisory Council for comments on earlier versions of the project. The project described in this publication was partially supported by the University of Rochester CTSA award number UL1 TR002001 from the National Center for Advancing Translational Sciences of the National Institutes of Health. The content is solely the responsibility of the authors and does not necessarily represent the official views of the National Institutes of Health."

"JH, JHW, AC, and MZ received funding through the University of Rochester CTSA award number UL1 TR002001 from the National Center for Advancing Translational Sciences of the National Institutes of Health. The funder played no role in the study design, data collection and analysis, decision to publish or preparation of the manuscript."

4. Please upload a copy of Supporting Information Figure/Table/etc. Fig S7-S10 and Table S8-S11 which you refer to in your text on page 19.

Additional Editor Comments:

Dear Dr. Jonathan, Herington,

Thank you for allowing us to review your manuscript: PONE-D-24-08864 "Investigating Ethical Tradeoffs in Crisis Standards of Care through Simulation of Ventilator Allocation Protocols." We require minor revisions and please submit as soon as possible. If you are not able to see the reviewers' comments, do not hesitate to contact me.

Best regards,

Ronaldo Go MD

Reviewers' comments:

Reviewer's Responses to Questions

**Comments to the Author**

1. Is the manuscript technically sound, and do the data support the conclusions?

Reviewer #1: Yes

Reviewer #2: Yes

Reviewer #3: Yes

2. Has the statistical analysis been performed appropriately and rigorously? 

Reviewer #1: Yes

Reviewer #2: Yes

Reviewer #3: Yes

3. Have the authors made all data underlying the findings in their manuscript fully available?

Reviewer #1: Yes

Reviewer #2: Yes

Reviewer #3: Yes

4. Is the manuscript presented in an intelligible fashion and written in standard English?

Reviewer #1: Yes

Reviewer #2: Yes

Reviewer #3: Yes

5. Review Comments to the Author

Reviewer #1: In this excellent and well-written study, Herrington et. al perform a monte carlo simulation of ventilator allocation under crisis standards of care. They compare the results of various published triage protocols, finding that age-based triage vastly outperforms SOFA-tiers in terms of lives and life-years saved. The analysis adds significantly to the prior literature and has important findings for refining crisis standards of care for future disasters. The simulation approach is very clever in handling the various capacity levels and the visual representations of the simulation results are outstanding.

I believe the following comments will strengthen the manuscript:

1. More details on the SOFA score calculation should be added to the methods. Were the SOFA score components calculated automatically by the EHR software or by the investigators from the underlying EHR data? Was the original 1997 definition of SOFA used?

2. When exactly was the SOFA score calculated? Was the worst value of each SOFA component in the first 24 hours following the initiation of ventilation used to calculate the total SOFA score? The discussion says the study only used “SOFA scores collected at point of intubation”, which implies this calculation was happening using data from the 24 hours prior to intubation. This needs to be clarified in the methods.

3. 320 patients were excluded because they lacked lab results for SOFA, but earlier missing SOFA values were “assumed to be normal”.

4. What does “an unclear pattern of admissions and intubations (n = 332)” mean exactly?

5. I would not recommend referring to the SOFA plus comorbidity with age tie-breaker as “the Bhavani protocol” throughout the paper. This protocol was not explicitly endorsed by Bhavani or the other authors of the manuscript and seems to have been designed to represent several different state plans (e.g., Maryland, Pennsylvania).

6. What was the etiology of respiratory failure requiring mechanical ventilation in the 66% of patients who did not have COVID? This information should be added to Table 1.

7. Related to the previous comment, the patient population during a crisis scenario would likely differ from the general population who required mechanical ventilation from 4/2020- 5/2021. For example, during an overwhelming COVID-19 surge, patients who required mechanical ventilation following an elective surgery would not be included and the cohort would likely be mostly COVID +. I recommend the authors run sensitivity analyses with sub-populations for specific crisis scenarios (e.g. 90% COVID + and no elective surgery patients).

8. Did the authors race-adjust life-expectancy and life-years in Table 4? I would argue this choice normalizes racial disparities in life-expectancy which hopefully will improve over time with efforts to address structural inequity in healthcare. So I recommend these statistics be re-calculated for age-alone.

9. The updated Colorado guidelines currently include a correction to the renal SOFA score component in an attempt to debias SOFA. I recommended adding this to the simulation results.

Minor:

In the abstract, would revise to the following: For a capacity of 1 bed per 2 patients, ranking by age bands saves approximately 28.7 lives and 3408 life-years per thousand patients, while ranking by Sequential Organ Failure Assessment

20 (SOFA) bands saved 13.2 lives and 416 life-years per patient (**statistical testing**)

Reviewer #2: I congratulate you on a well done study with important findings. AS pointed out, in crisis mode, many institutions and governing bodies have suggested using allocation algorithms employing SOFA scores. Although simple to calculate and readily available, the SOFA score does not contain parameters for age. As your paper has pointed out, age is clearly a very important factor to consider in saving lives and life years. Also as you pointed out in the discussion, there are many political and legal issues with using age concerning discrimination.

Although the most optimal way to stratify patients during periods of CSC remains unclear, this paper advocates that SOFA alone may not be enough, and that age should be included in some way. It's important to note that the Colorado system utilizes the Charlson comorbidity Index, and this index contains age as one of the components in its calculation with significant weight.

I also had a question about the relatively low percentage of COVID cases. During this crisis time frame, most hospitals were overwhelmed with COVID cases, typically shutting down ORs except for emergency cases, etc. It seems odd that only 34% of the critically ill cases would be secondary to COVID. The overall survival between COVID and non COVID was similar, but was there a further subgroup analysis?

I appreciate that breaking down SOFA scores into tiers or bands has its drawbacks and is likely a significant limitation with the NY score using only three bands. This will also lead to more tiebreaks which are done by lottery.

Lastly I agree there seems to be a population bias which is heavily affecting the racial outcomes. This part of the paper was difficult to explain. I encourage you to do the regional analysis as you suggested.

One small point.

Methods section Line 64: You state the implementation of five proposed ventilator allocation protocols: You then describe six protocols.

Reviewer #3: This study evaluates 5 proposed CSC (Crisis Standards of Care) protocols for rationing ventilators during the COVID19 pandemic for adults that required a ventilator between 4/2020 and 5/2021 in a Monte Carlo simulation model of ventilator allocation. The strengths of this study include the inclusion of non-COVID patients with respiratory failure requiring mechanical ventilation proving a more realistic dataset compared to other studies.

The limitations are well documented by the authors and include that it is a single center study, the non-Hispanic black and Hispanic patients are a much smaller sample than the white patients, many non-Hispanic black and Hispanic patients may have been triaged to a different hospital within the same area, and there was no accounting for DNR code status.

Similar to other studies are the findings that allocation protocol favoring younger age saved the most lives and added the most “life-years” and SOFA based protocol saved fewer lives than age allocated. What is different from other studies is the observation that whites experienced significantly lower allocation rates, lives saved, and life-years saved with the implementation of all the protocols except two: NY SOFA and lottery.

Since age has a strong relationship with survival from COVID, and the black and Hispanic patients were younger, this may partly explain why they fared better than the older white patients. But as pointed out bias in the distribution of patients between two different metropolitan hospitals that serve particular patient populations may also result in these differences.

The authors are to be commended for their thorough analytic approach, clear presentation of the statistics, thoughtful interpretation of the results, and the provided link to the code used to perform the analysis.

Typo:

“agre” = age

Table S8: Survival by Agre Group at 50% capacity.

6. PLOS authors have the option to publish the peer review history of their article (what does this mean?). If published, this will include your full peer review and any attached files.

Reviewer #1: **Yes: **William F Parker

Reviewer #2: No

Reviewer #3: No

---

## [Author Response · Author response to Decision Letter 0]

9 Jul 2024

Response to reviewers is included as a .docx file alongside manuscript.

---

## [Decision Letter · Decision Letter 1]

25 Jul 2024

Investigating Ethical Tradeoffs in Crisis Standards of Care through Simulation of Ventilator Allocation Protocols

PONE-D-24-08864R1

Dear Dr. Herington,

We’re pleased to inform you that your manuscript has been judged scientifically suitable for publication and will be formally accepted for publication once it meets all outstanding technical requirements.

Kind regards,

Ronaldo Go, MD

Academic Editor

PLOS ONE

Additional Editor Comments (optional):

Reviewers' comments:

Reviewer's Responses to Questions

**Comments to the Author**

1. If the authors have adequately addressed your comments raised in a previous round of review and you feel that this manuscript is now acceptable for publication, you may indicate that here to bypass the “Comments to the Author” section, enter your conflict of interest statement in the “Confidential to Editor” section, and submit your "Accept" recommendation.

Reviewer #1: All comments have been addressed

Reviewer #2: All comments have been addressed

2. Is the manuscript technically sound, and do the data support the conclusions?

Reviewer #1: Yes

Reviewer #2: Yes

3. Has the statistical analysis been performed appropriately and rigorously? 

Reviewer #1: Yes

Reviewer #2: Yes

4. Have the authors made all data underlying the findings in their manuscript fully available?

Reviewer #1: Yes

Reviewer #2: Yes

5. Is the manuscript presented in an intelligible fashion and written in standard English?

Reviewer #1: Yes

Reviewer #2: Yes

6. Review Comments to the Author

Reviewer #1: (No Response)

Reviewer #2: I believe this should be accepted. It is well written and a well performed analysis. I believe the conclusions are valid.

All of my concerns have been addressed. I look forward to seeing this article published. Thank you for allowing me to review this article.

7. PLOS authors have the option to publish the peer review history of their article (what does this mean?). If published, this will include your full peer review and any attached files.

Reviewer #1: **Yes: **William F Parker

Reviewer #2: **Yes: **Keith M. Rose, MD

---

## [Editor Report · Acceptance letter]

31 Jul 2024

PONE-D-24-08864R1 

PLOS ONE

Dear Dr. Herington, 

I'm pleased to inform you that your manuscript has been deemed suitable for publication in PLOS ONE. Congratulations! Your manuscript is now being handed over to our production team.

Kind regards, 

on behalf of

Dr. Ronaldo Go 

Academic Editor

PLOS ONE